EMBO
Molecular Medicine

# DNA methylation changes during acute COVID-19 are associated with long-term transcriptional dysregulation in patients' airway epithelial cells

Marey Messingschlager [1,2,12], Sebastian D Mackowiak[3,12], Maria Theresa Voelker [4,12], Matthias Bieg[3], Jennifer Loske [1,2], Robert Lorenz Chua[3], Johannes Liebig[3], Sören Lukassen[3], Loreen Thürmann[1], Anke Seegebarth[1], Sven Twardziok[3], Daria Doncevic[5], Carl Herrmann [5], Stephan Lorenz[6], Sven Klages [6], Fridolin Steinbeis[7], Martin Witzenrath[7,8], Florian Kurth[7,9,10], Christian Conrad [3], Leif E Sander[7,8], Naveed Ishaque [3,12], Roland Eils[3,5,8,11,12], Irina Lehmann [1,8,12], Sven Laudi [4,12] & Saskia Trump [1,12] ✉

## Abstract

**Molecular changes underlying the persistent health effects after SARS-CoV-2 infection remain poorly understood. To discern the gene regulatory landscape in the upper respiratory tract of COVID-19 patients, we performed enzymatic DNA methylome and single-cell RNA sequencing in nasal cells of COVID-19 patients ($n = 19$, scRNA-seq $n = 14$) and controls ($n = 14$, scRNA-seq $n = 10$). In addition, we resampled a subset of these patients for transcriptome analyses at 3 ($n = 7$) and 12 months ($n = 5$) post infection and followed the expression of differentially regulated genes over time. Genome-wide DNA methylation analysis revealed 3112 differentially methylated regions between COVID-19 patients and controls. Hypomethylated regions affected immune regulatory genes, while hypermethylated regions were associated with genes governing ciliary function. These genes were not only downregulated in the acute phase of the disease but sustained repressed up to 12 months post infection in ciliated cells. Validation in an independent cohort collected 6 months post infection ($n = 15$) indicated symptom-dependent transcriptional repression of ciliary genes. We therefore propose that hypermethylation observed in the acute phase may exert a long-term effect on gene expression, possibly contributing to post-acute COVID-19 sequelae.**

**Keywords** Whole-genome DNA Methylation Sequencing; scRNA-seq; COVID-19; Post-COVID-19 Condition; Nasal Mucosa
**Subject Categories** Chromatin, Transcription & Genomics; Respiratory System

## Introduction

To date, the coronavirus disease 2019 (COVID-19) pandemic has caused more than 7 million deaths worldwide (Our World in Data, 2024). Despite a high proportion of immunized individuals due to infection, vaccination, or both, thousands continue to be (re-) infected every day, albeit with declining mortality and morbidity (World Health Organization, 2023). While the threats of acute infection diminished over the last years, persistent post-COVID symptoms emerged as a new, but relevant problem for the healthcare system (Katz et al, 2023). In view of the persistently high numbers of ongoing infections and the associated risk of enduring post-COVID symptoms that exist with every new infection (Bowe et al, 2022), it remains a major goal to comprehensively understand the mechanisms of infections with severe acute respiratory syndrome coronavirus type 2 (SARS-CoV-2) and resulting long-term consequences.

Although SARS-CoV-2 infections affect multiple organs and elicit a systemic response (Petersen et al, 2022), the nasopharynx constitutes the initial site of SARS-CoV-2 infection and replication (Wolfel et al, 2020). Upon infection, the airway epithelium releases interferons and pro-inflammatory cytokines (Park and Iwasaki, 2020), triggering the infiltration of immune cells, with severe COVID-19 often being characterized by excessive recruitment and activation of myeloid cells, in particular neutrophils and macrophages (Chua et al, 2020; Voiriot et al, 2022; Ziegler et al, 2021).

[1]Berlin Institute of Health at Charité—Universitätsmedizin Berlin, Center of Digital Health, Molecular Epidemiology Unit, Berlin, Germany. [2]Freie Universität Berlin, Institute of Biology, Berlin, Germany. [3]Berlin Institute of Health at Charité—Universitätsmedizin Berlin, Center of Digital Health, Berlin, Germany. [4]Department of Anesthesiology and Intensive Care, University Hospital Leipzig, Leipzig, Germany. [5]Health Data Science Unit, Heidelberg University Hospital and BioQuant, University of Heidelberg, Heidelberg, Germany. [6]Max Planck Institute for Molecular Genetics, Berlin, Germany. [7]Charité—Universitätsmedizin Berlin, corporate member of Freie Universität Berlin and Humboldt-Universität zu Berlin, Department of Infectious Diseases and Respiratory Medicine, Berlin, Germany. [8]German Center for Lung Research (DZL), Giessen, Germany. [9]Department of Tropical Medicine, Bernhard Nocht Institute for Tropical Medicine, Hamburg, Germany. [10]Department of Medicine, University Medical Center, Hamburg-Eppendorf, Hamburg, Germany. [11]Department of Mathematics and Computer Science, Freie Universität Berlin, Berlin, Germany. [12]These authors contributed equally: Marey Messingschlager, Sebastian D Mackowiak, Maria Theresa Voelker, Naveed Ishaque, Roland Eils, Irina Lehmann, Sven Laudi, Saskia Trump. ✉E-mail: saskia.trump@bih-charite.de

While the transcriptome and proteome of the nasal mucosa during SARS-CoV-2 infection have been studied comprehensively (Chua et al, 2020; Trump et al, 2021; Vanderboom et al, 2021; Ziegler et al, 2021), the underlying regulatory changes are still largely unexplored. DNA methylation represents one of the main epigenetic mechanisms by which cells can regulate and stabilize gene expression. As DNA methylation patterns are transmitted across cell divisions, changes in the DNA methylome can lead to the heritable reprogramming of cellular transcriptional programs (Mattei et al, 2022). As such, DNA methylation becomes particularly interesting with regard to health impairments that persist subsequent to infection, also known as post-COVID-19 condition (PCC) or post-acute sequelae of COVID-19. PCC occurs in ~10% of infected individuals, with higher rates in hospitalized or unvaccinated individuals, and encompasses a variety of symptoms including respiratory, cardiovascular, or cognitive impairments (Davis et al, 2023). While many of these symptoms can resolve within a few months to one year, some, such as dyspnea or the loss of smell, have the tendency to persist longer (Bowe et al, 2023). Current studies on DNA methylation changes in COVID-19 almost exclusively have been limited to blood-derived samples (Balnis et al, 2021; Bernardes et al, 2020; Konigsberg et al, 2021), thereby excluding the epithelial cells of the nasal mucosa which may be of relevance with regard to respiratory post-acute sequelae.

To gain insight into the upper airway gene regulatory landscape during SARS-CoV-2 infection, we performed whole-genome enzymatic DNA methylation sequencing (mDNA-seq) along with single-cell RNA sequencing (scRNA-seq) in a cohort of 33 individuals. By combining methylome with scRNA data, we were able to assess the impact of the methylation changes on gene expression within the same sample on single-cell resolution. In addition, we resampled a subset of patients for transcriptome analyses at 3 and 12 months post infection (total $n = 12$), thereby allowing us to track the expression of differentially regulated genes over time. Lastly, the observed changes in gene expression were verified and further refined using scRNA-seq data from an independent cohort collected 6 months post infection ($n = 6$ persisting respiratory symptoms, $n = 9$ no respiratory symptoms).

# Results

## DNA methylation & scRNA sequencing cohort

The analysis encompassed mDNA-seq data obtained from nasopharyngeal cells of 19 COVID-19 patients (21–76 years; 16 males, 3 females) and 14 controls (24–73 years; 7 males, 7 females) (Table 1; EV1). In addition, for the majority of these samples (24 out of 33), matched single-cell transcriptome data were concurrently analyzed (Fig. 1A). From 10 COVID-19 patients, follow-up samples for scRNA-seq were obtained 3 months ($n = 7$) and 12 months after infection ($n = 5$). COVID-19 patients in both the main cohort (mDNA-seq) as well as the subcohort with mDNA-seq and scRNA-seq data tended to contain a larger proportion of males with a higher median age compared to the control group (Table 1). mDNA-seq data had a median genome-wide coverage of 62.3× across all samples (minimum 46.5×, Table EV2). The integrated scRNA-seq data comprised 82,365 cells assigned to 30 different cell types and states (Appendix Figs. S1 and S2).

## Characteristics of differentially methylated regions and target genes

To identify differences in the upper airway methylome between SARS-CoV-2 patients and controls, we determined age-, sex-, and cell proportion-adjusted differentially methylated regions (DMRs). The differential methylation analysis revealed a total of 3112 DMRs distributed across all autosomes, with a high proportion of DMRs (81.2%) that showed significantly higher mean methylation in SARS-CoV-2-positive individuals (i.e., hypermethylated, Fig. 1B, Dataset EV1). As expected within lowly methylated regions (LMRs) (Burger et al, 2013), a large proportion of DMRs overlapped with regulatory elements, such as enhancers or promoters (64.2%, Fig. 1C). The methylation quantitative trait loci (meQTL) analysis indicated that the majority of hypermethylated DMRs (hyperDMRs) were non-genotype-associated (ngDMRs, 65.7%), while hypomethylated DMRs (hypoDMRs) contained a higher proportion of gDMRs (60.8%, Fig. 1D). We identified a total of 5513 DMR target genes (Dataset EV1) and further examined these in order to understand the biological implications resulting from the methylation differences. Comparing the expression of DMR target genes between SARS-CoV-2 infected individuals and controls, we found that DMR target genes were rather upregulated in immune cells (59.8% of all 1859 differentially expressed genes [DEGs]), whereas the vast majority of epithelial DEGs were downregulated in COVID-19 patients compared to controls (91.9% of 2829 DEGs, Fig. 1E; Appendix Fig. S3; Dataset EV2).

## Hypomethylated regions affect genes involved in immune cell recruitment

Gene Ontology (GO) pathway enrichment showed that several genes derived from the cluster of hypoDMRs were involved in macrophage migration and cell chemotaxis, driven by a set of genes encoding chemokine receptors (Fig. 2A; Table EV3). These genes were derived from one ngDMR that overlapped with an enhancer from the GeneHancer database (GH03J046088) which had prior experimental evidence of interactions with the cluster of chemokine receptors on chr3p21.31 (Fig. 2B; Dataset EV1). As chemokine receptors are expressed on a variety of immune cells (Sobolik-Delmaire et al, 2013) and interactions between enhancers and their target genes are known to vary between different cell types, we used existing promoter capture Hi-C data from primary blood-derived leukocytes (Javierre et al, 2016) to identify cell type-specific interactions between the enhancer and its target genes (Fig. 2C). In line with the GeneHancer database, the Hi-C data showed several links between the enhancer and surrounding genes for the different immune cell populations, particularly for the promoter regions of CCR1, CCR2, and CCR5, with the highest number of interactions in monocytes, macrophages, and CD8-positive T cells. To further narrow down the cell types and genes affected by the hypomethylation, we used our sample-matched scRNA-seq data and correlated average methylation levels within the DMR with pseudo-bulked average expression of all enhancer target genes in the different immune cell types (Fig. 2C; Dataset EV3). The analysis showed that methylation levels of the hypoDMR were most frequently inversely correlated with the expression of CCR1; besides CD8-positive T cells and myeloid dendritic cells, significant correlation for CCR1 was detected in all three macrophage populations (monocyte-derived, non-resident (Fig. 2D) and resident macrophages), suggesting that the hypomethylation on chr3p21.31 particularly drives the upregulation of CCR1 in macrophages.

**Table 1. Distribution of cohort characteristics between SARS-CoV-2-positive patients and controls in the full (mDNA-seq) cohort and the scRNA-seq subcohort.**

| | mDNA-seq cohort (n = 33) | | |
|---|---|---|---|
| | **SARS-CoV-2 (n = 19)** | **Controls (n = 14)** | |
| Female/male, n (%) | 3 (15.8)/16 (84.2) | 7 (50.0)/7 (50.0) | |
| Age (years), median (range) | 57 (21–76) | 38.5 (24–73) | |
| BMI, median (range) | 25.1 (20.0–36.2)[a] | 28.1 (19.4–41.9) | |
| Cardiovascular disease, n (%) | 9 (47.3) | 4 (28.6) | |
| Pre-existing respiratory disease, n (%)[b] | 3 (15.8) | 1 (7.1) | |
| Diabetes, n (%) | 4 (21.1) | – | |
| COVID-19 disease severity[c] | | | |
| mild or moderate, n (%) | 8 (42.1) | N/A | |
| severe or critical, n (%) | 11 (57.9) | N/A | |
| Number of days hospitalized, median (range) | 18 (7–88)[d] | N/A | |
| | **Subcohort with mDNA-seq and scRNA-seq data (n = 24)** | | |
| | **SARS-CoV-2 (n = 14)** | **Controls (n = 10)** | |
| Female/male, n (%) | 2 (14.3)/12 (85.7) | 4 (40.0)/6 (40.0) | |
| Age (years), median (range) | 60 (21–76) | 35 (24–73) | |
| BMI, median (range) | 27.1 (20.0–36.2)[a] | 26.2 (19.4–41.9) | |
| Cardiovascular disease, n (%) | 6 (42.9) | 3 (30.0) | |
| Pre-existing respiratory disease, n (%)[b] | 3 (21.4) | – | |
| Diabetes, n (%) | 4 (28.6) | – | |
| COVID-19 disease severity[c] | | | |
| Mild or moderate, n (%) | 5 (35.7) | N/A | |
| Severe or critical, n (%) | 9 (64.3) | N/A | |
| Number of days hospitalized, median (range) | 18 (7–88)[e] | N/A | |

[a]BMI info is missing for one SARS-CoV-2-positive patient.
[b]Asthma or COPD.
[c]COVID-19 disease severity based on WHO classification (https://www.who.int/docs/default-source/coronaviruse/who-china-joint-mission-on-covid-19-final-report.pdf).
[d]Two patients deceased.
[e]One patient deceased.

## Hypermethylation is associated with repression of genes involved in ciliary function

To understand the impact of hypermethylation on epithelial cells, we performed a gene set enrichment analysis based on the differential expression of the hyperDMR target genes in all epithelial cells (Dataset EV4). In line with the hypermethylation, GO pathway enrichment showed that the vast majority of the enriched pathway genes were suppressed in epithelial cells of COVID-19 patients, while only a small proportion were upregulated (Fig. 3A; Table EV4). The suppressed pathway genes were implicated in cytoskeletal processes and in particular ciliary

function; including genes involved in the generation of motile cilia (*RSPH9, DNAH3,* and *DNAH5*) (Tilley et al, 2015), genes that are part of the ciliary protein trafficking machinery (*IFT122, IFT46*) (Tilley et al, 2015), as well as the ciliogenesis regulating transcription factor *RFX3* (El Zein et al, 2009) and its downstream effectors *ALMS1* (Purvis et al, 2010), *TEKT1* and *TEKT2* (Didon et al, 2013). In order to determine how the observed transcriptional repression affects the different epithelial cell subtypes (basal, secretory, and ciliated cells, see Appendix Fig. S2), we calculated a score based on the average expression of all suppressed pathway genes per cell, named *pathways score* (n = 323 genes, Dataset EV5, Fig. 3A, lower panel). In negative controls, ciliated cells had the highest *pathways score*, followed by secretory and basal cells, the progenitors of ciliated cells (Davis and Wypych, 2021) (Fig. 3B, left panel). In contrast, in the acute phase of COVID-19, the *pathways score* was reduced across all epithelial cells (Fig. 3B, right panel, Table EV5), in conjunction with a general loss of epithelial cells and increased proportions of immune cells, mostly of neutrophils and non-resident macrophages (Appendix Fig. S4; Table EV6).

## Cell composition changes in postinfection follow-up samples

The majority of patients subjected to follow-up sampling at 3 and 12 months post infection (n = 10, Fig. 4A) exhibited persisting respiratory symptoms, such as exercise-induced dyspnea (n = 7/10, Tables EV1 and EV7, Appendix Fig. S5). Taking these samples into account, we followed the development of cellular proportions, and the *pathways score* over time. After the aforementioned decline in epithelial cell proportions during the acute phase of infection, the samples collected 3 months post infection still showed an elevated immune cell fraction, which returned to the level of controls after 12 months (Appendix Fig. S4). While the percentage of neutrophils remained elevated in the 3 month follow-up samples, non-resident macrophage numbers decreased more rapidly after infection. The proportion of epithelial cells gradually increased over time and recovered to the level of controls at 12 months post infection.

## Epithelial cells show differences in gene expression recovery

Following the reduction in the acute phase of infection, the *pathways score* increased gradually from the acute phase to 12 months post infection in basal and secretory cells. Although ciliated cells also showed a partial recovery of the *pathways score* in the 3-month follow-up samples, overall expression of constituent genes remained repressed at 12 months after infection (Fig. 4B; Table EV5). As our data structure did not allow for a fully longitudinal assessment, we stratified the cohort to compare the acute phase to the respective follow-up time point in a patient-matched manner (Appendix Fig. S6). The matched comparison supported the trends observed in the combined analysis, showing a repression of the majority of pathway score genes compared to controls at both short and long-term follow-up time points. To identify those *pathways score* genes that remained repressed in ciliated cells, we performed a differential expression analysis comparing the 12-month follow-up samples to controls (Dataset EV6). In line with the maintained repression of the *pathways score*, we found a large number of downregulated genes in ciliated cells (n = 106), while a much smaller number remained repressed in basal and secretory cells. Most of those

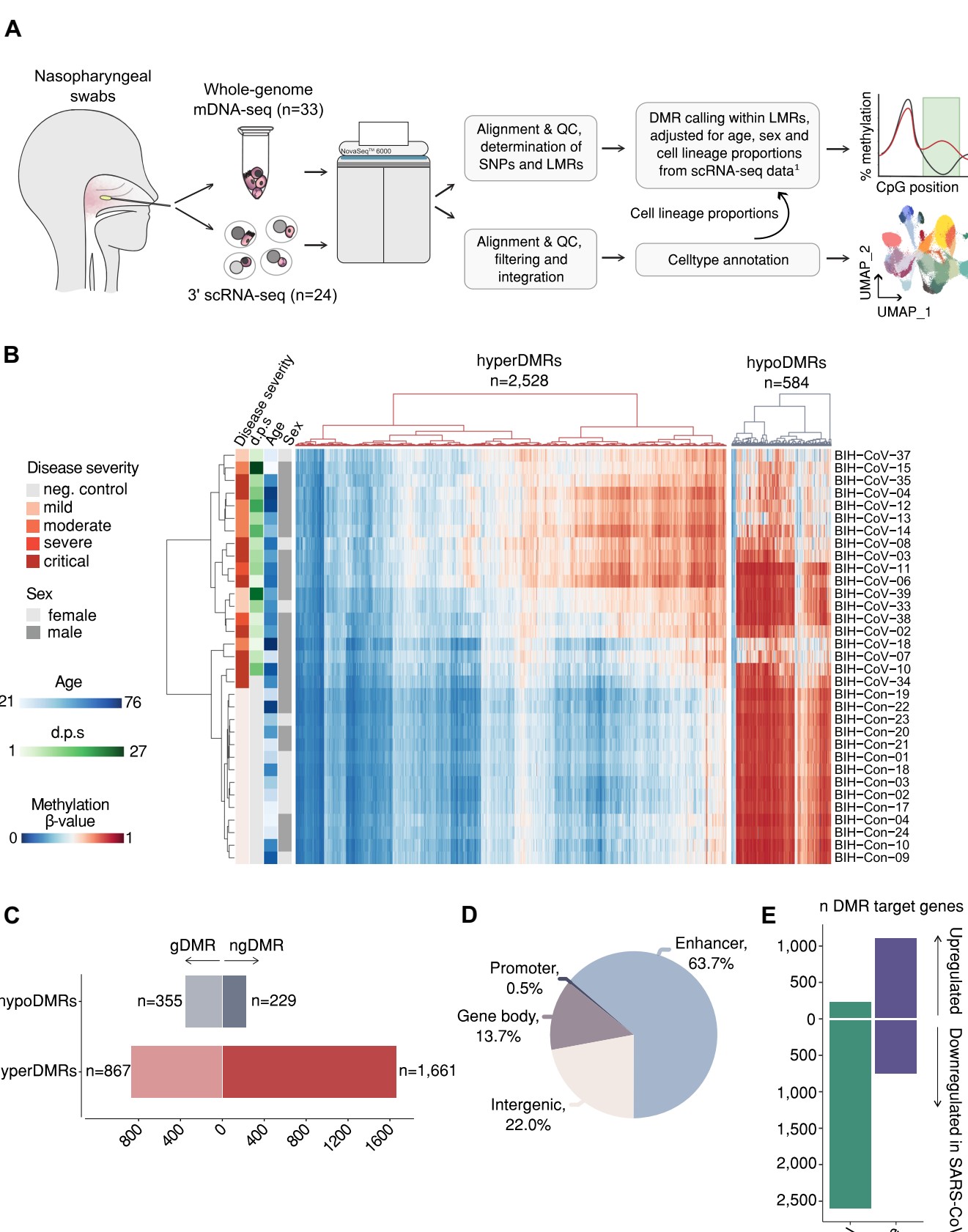

◀    **Figure 1.   Sequencing workflow and characteristics of DNA methylome data.**

(A) Sequencing workflow and analysis scheme. [1]For those samples, where scRNA-seq data were not available ($n = 9$), cell lineage proportions were inferred using EPISCORE (B) Heatmap representation of all differentially methylated regions (DMRs, $n = 3112$) comparing SARS-CoV-2-positive individuals ($n = 19$) to controls ($n = 14$), adjusted for age, sex, and cellular composition. Hyper- and hypomethylated DMRs were clustered separately. Heatmap colors encode mean methylation per DMR (columns) and sample (rows). d.p.s. days post start of symptoms. (C) Genomic features of DMRs. (D) Distribution of DMRs across hypo- and hypermethylated as well as genotype-associated (g) or non-genotype-associated (ng) DMRs. (E) Number of differentially expressed DMR target genes (DEGs) between SARS-CoV-2-positive patients ($n = 14$) vs. controls ($n = 10$) in all epithelial or immune cells combined (from Dataset EV2). Differential gene expression was calculated using the MAST test, adjusted for age, sex, and fraction of genes per cell. SNP single-nucleotide polymorphism, LMR lowly methylated region, DMR differentially methylated region.

genes that were downregulated in secretory cells, however, were also repressed in ciliated cells (Appendix Fig. S7a). On the contrary, especially in secretory cells, *pathways score* genes tended to be upregulated 12 months post infection and showed little overlap with those genes that remained repressed in ciliated cells (Appendix Fig. S7b). This indicates that it is distinct subsets of pathway genes in the different epithelial cell subtypes that either regained their transcriptional levels or remained repressed at 12 months after infection.

## Ciliated cells exhibit persistent gene suppression

To further assess the functions of repressed *pathways score* genes in ciliated cells, we conducted a gene network analysis (Fig. 4C; Table EV8), which showed that a large proportion of the persistently suppressed genes was involved in ciliogenesis, microtubule-based movement, and the regulation of protein localization, with many of the genes from the latter two modules having been described specifically in the context of the ciliary function, such as *DNAH3*, *DNAH5* (Tilley et al, 2015), *CCT3*, and *CCT8* (Seo et al, 2010). Also, the ciliogenesis regulator *RFX3* remained repressed in ciliated cells at 12 months post infection (Dataset EV6). Depicting the expression of the individual network genes confirmed that the overall suppression observed during the acute phase was sustained, albeit slightly increasing after 3 and 12 months post infection, but not yet reaching the level of controls (Fig. 4D; Appendix Fig. S8). In line, analysis of transcription factor binding motifs in those hyperDMRs whose target genes remained repressed 12 months post infection in ciliated cells revealed enrichment for transcription factors involved in transcriptional repression, cell differentiation, and reprogramming, such as HEY2, GLIS1, and REST (Table EV9) (Jetten et al, 2022; Jin et al, 2023; Weber et al, 2014). Notably, in contrast to the persistently reduced expression of hyperDMRs target genes, no long-term impact on gene expression of the hypoDMR-related genes described in Fig. 2 was observed (Appendix Fig. S9; Table EV10).

## Independent follow-up cohort: symptom-linked ciliary transcriptional perturbation in post-COVID-19

While the mDNA-seq and scRNA-seq cohort allowed us to track expression levels across time points, the follow-up cohort was unbalanced insofar as the samples were composed mainly of male donors with severe acute disease. To verify our observations in an independent cohort, we included scRNA-seq data from samples collected 6 months post-SARS-CoV-2 infection from individuals with mild acute disease and a higher proportion of females ($n = 15$, Table EV11, Appendix Fig. S5). In line with the scRNA-seq data from the samples collected 3 and 12 months post infection, the pathways score levels were significantly reduced compared to controls at 6 months post infection (Appendix Fig. S10a). As this

independent cohort was composed of $n = 6$ donors that reported persisting respiratory symptoms and $n = 9$ without persisting symptoms, we were further able to examine symptom-dependent differences in the *pathways score*. All patients experiencing respiratory symptoms reported a reduction in physical fitness compared to their pre-COVID-19 state and did not experience dyspnea symptoms prior to infection. In ciliated cells, *pathways score* levels were significantly reduced in patients with persisting respiratory symptoms vs. those without (multiple linear regression, adjusted for age, sex, and fraction of genes per cell, coefficient = $-0.002$ (95% CI $-0.004$ to $-0.001$), $P$ value $< 0.001$). These differences were most pronounced in the differentiated ciliated cell population (Appendix Fig. S10b). A detailed examination of the *pathways score* genes through differential gene expression analysis revealed that genes involved in ciliogenesis and the coordination of ciliary beating, such as *RFX3*, *EZR* and *DNAH3* (El Zein et al, 2009; Kawaguchi et al, 2022; Tilley et al, 2015), were among the most strongly repressed genes in patients with persisting respiratory symptoms (Appendix Fig. S10c,d; Dataset EV7).

## Discussion

In this study, we conducted the first characterization of the genome-wide DNA methylome together with the single-cell transcriptome in the nasal mucosa of adult COVID-19 patients. By collecting single-cell gene expression data not only during the acute phase, but also 3, 6 and 12 months post infection, we investigated both immediate and persistent changes in gene expression in the upper airways.

Previous studies on DNA methylation changes in COVID-19 have primarily focused on array-based investigations of blood-derived samples (Balnis et al, 2021; Bernardes et al, 2020; Konigsberg et al, 2021), that are limited to a subset of 3.5% of all CpGs in the human genome (Pidsley et al, 2016). In contrast, we assessed global epigenetic alterations using mDNA-seq in cells from the nasal mucosa, the primary site of SARS-CoV-2 infection and replication (Wolfel et al, 2020). The combination of methylome and single-cell transcriptome data further allowed us to infer cell type-specific effects related to the methylation changes.

We identified a hypoDMR within the cluster of chemokine receptors on chr3p21.3 that associated with the upregulation of *CCR1*, particularly in macrophages. This DMR was in close proximity to the genetic risk locus linked to severe COVID-19 (Zeberg and Paabo, 2020), aligning with previous research indicating that upregulation of *CCR1* in macrophages is related to severe COVID-19 outcomes (Chua et al, 2020; Trump et al, 2021). It has been suggested that active chromatin states in this risk locus govern the expression of *CCR1* particularly in macrophages

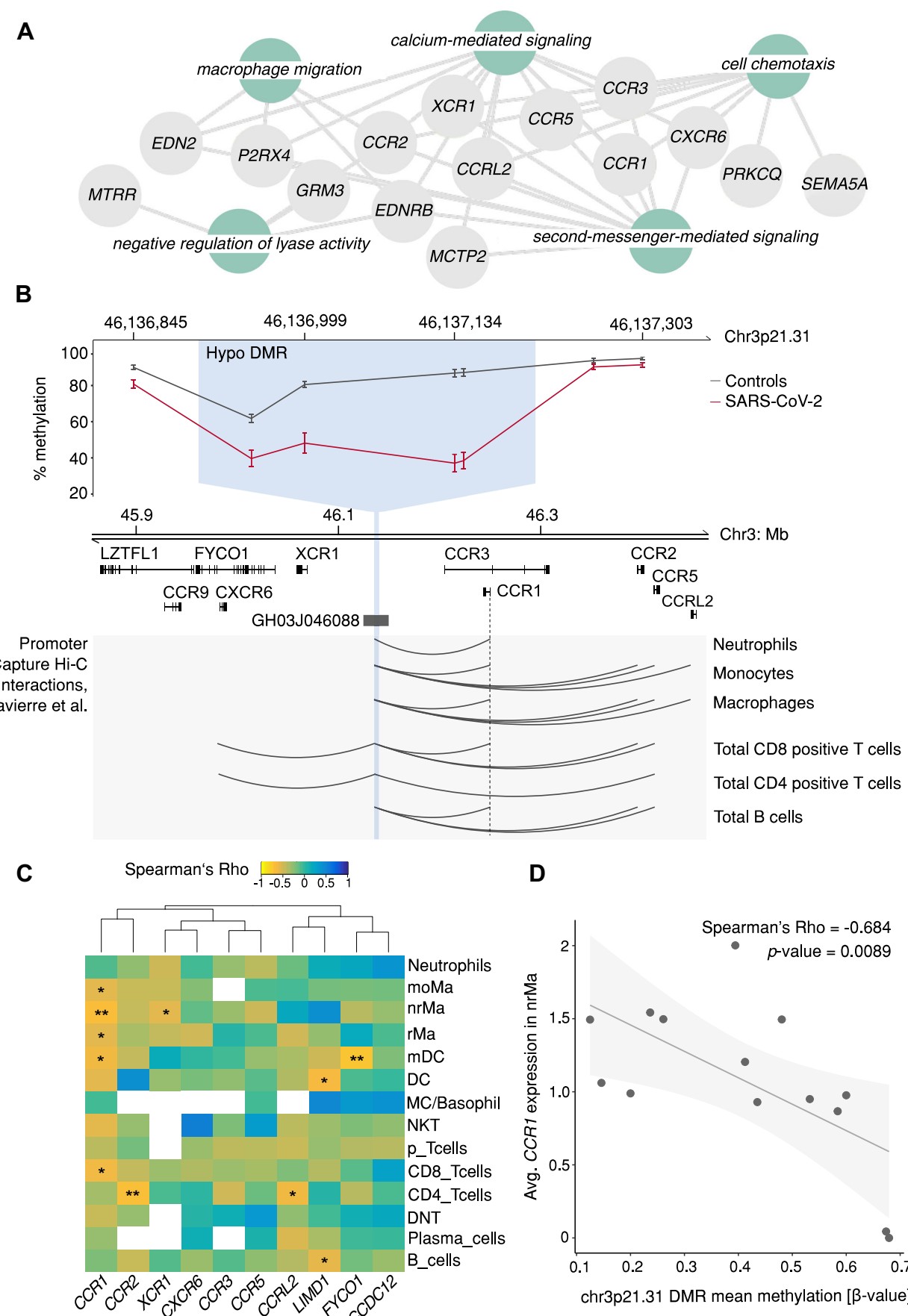

◄ **Figure 2. Hypomethylated regions affect genes involved in immune cell recruitment.**

(A) Network analysis of hypoDMR target genes based on Enrichr-KG. (B) Genomic view of one hypoDMR located in the cluster of chemokine receptors on chr3p21.31. The line plot on the top displays mean methylation values with standard errors per group (SARS-CoV-2-positive $n = 14$, controls $n = 10$), with the DMR indicated by blue shading. The DMR overlapped with an enhancer (GH03J046088) whose interactions with target gene promoters in primary immune cell types are shown in the bottom panel (Promoter capture Hi-C data from Javierre et al, 2016). (C) Correlation between mean methylation levels at the hypoDMR and average expression of enhancer target genes in immune cells of SARS-CoV-2-positive patients ($n = 14$). Heatmap colors indicate Spearman correlation coefficients. Missing correlation coefficient = gene not expressed by respective cell type. *$P < 0.05$, **$P < 0.01$. (D) Example scatterplot depicting the relationship between DNA methylation and *CCR1* expression in non-resident macrophages in SARS-CoV-2-positive samples. DMR differentially methylated region, moMa monocyte-derived macrophages, nrMa non-resident macrophages, rMa resident macrophages, mDC myeloid dendritic cells, DC dendritic cells, MC/Basophil mast cells or basophils, NKT natural killer T cells, p_Tcells proliferating T cells, DNT double negative T cells.

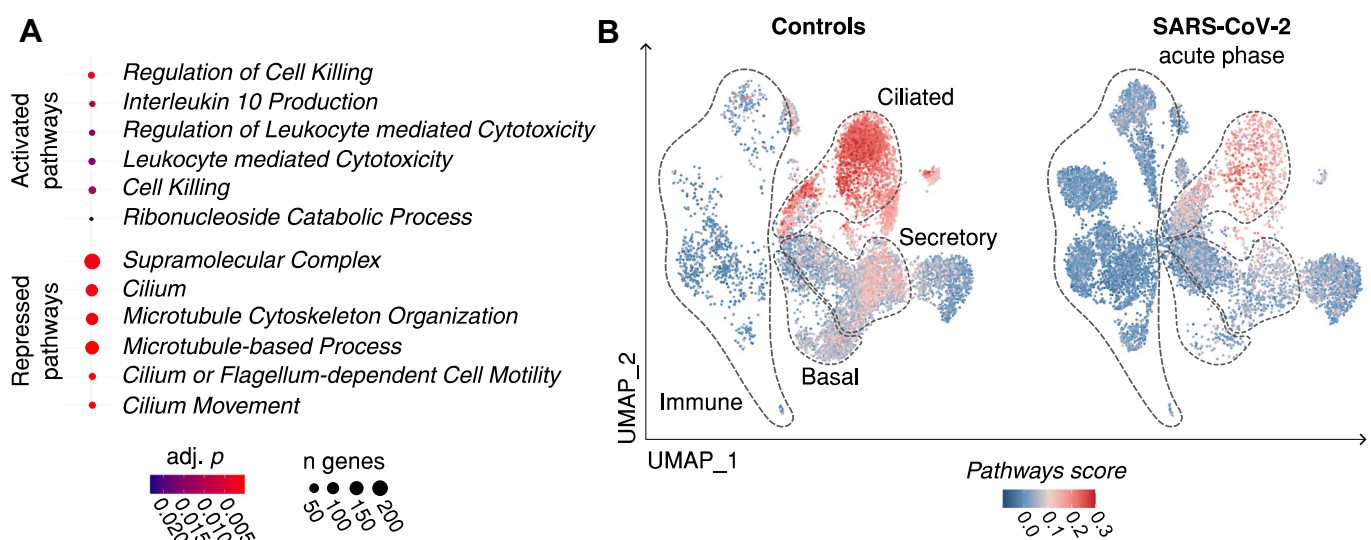

**Figure 3. Transcriptional repression of ciliary genes during SARS-CoV-2 infection.**

(A) Gene set enrichment analysis was performed based on the expression of hyperDMR target genes in epithelial cells (Dataset EV4). Activated and repressed pathways are shown. (B) The expression of all genes contained within the repressed pathways ($n = 323$, from (A), lower panel) is represented by the *pathways score*. *Pathways score* levels are shown for nasal cells from controls (left, $n = 10$) and SARS-CoV-2-positive patients (right, $n = 14$). For comparability, cells were downsampled to 9000 cells per group. DMR differentially methylated region.

(Stikker et al, 2022). The observed inverse correlation between DNA methylation and *CCR1* expression in macrophages, implies that DNA hypomethylation adds an additional layer of regulation to *CCR1* induction and macrophage recruitment, as it has been previously described in the context of COVID-19 (Chua et al, 2020; Stikker et al, 2022; Trump et al, 2021).

Previous studies on DNA methylation in blood of COVID-19 patients have identified a large proportion of hypomethylated regions, aligning with an activated immune response (Balnis et al, 2021; Bernardes et al, 2020). Despite the critical role immune cells play in antiviral defense, the primary targets for infection are epithelial cells, in particular ciliated cells (Ahn et al, 2021; Lukassen et al, 2020; Wu et al, 2023). In the acute phase of COVID-19, there is a significant depletion of ciliated cells, necessitating extensive regeneration of the epithelial layer (Robinot et al, 2021; Schreiner et al, 2022; Ziegler et al, 2021). Not only a loss of cells but also functional impairment of the epithelium has been described due to SARS-CoV-2 infection, including deciliation and diminished ciliary beat frequency (Li et al, 2023; Vijaykumar et al, 2023). Correspondingly, impaired upper airway mucociliary clearance has been described in dyspneic COVID-19 patients (Pezato et al, 2023). In line with these reports, our study shows pronounced DNA

hypermethylation in nasal epithelial cells of COVID-19 patients, associated with repression of genes involved in epithelial cell function.

Differences in gene expression can result from heritable changes in the epigenetic landscape that promote reprogramming of the cellular response and phenotype. Therefore, methylation changes in epithelial cells initiated in the acute phase of COVID-19 may mediate the observed long-lasting repression of hypermethylation-related genes involved in ciliogenesis up to 12 months post infection. This notion is supported by earlier animal models and in vitro studies that showed prolonged effects on ciliogenesis-related genes following SARS-CoV-2 infection (Robinot et al, 2021; Schreiner et al, 2022). Although we lack DNA methylation data for the follow-up samples, several other studies have shown that DNA methylation patterns can persist long after SARS-CoV-2 infection (Balnis et al, 2023; Balnis et al, 2022; Huoman et al, 2022). The fact that secretory cells, the progenitors of ciliated cells, showed similar sustained expression changes points towards an impaired differentiation potential. In agreement with this, altered epithelial differentiation and a reduced proportion of ciliated cells have been observed in patients with severe post-COVID sequelae (Fähnrich et al, 2024). It is therefore conceivable that the transcriptional programs of newly differentiated ciliated cells may remain altered due

to propagated epigenetic perturbations induced during the acute infection phase, also affecting progenitor cells. Given the role of DNA methylation in persistently altering transcriptional responses, we posit that an inherited reprogramming of epithelial cells may underlie COVID-19 sequelae like dyspnea or olfactory dysfunction.

While this study provides valuable insights into epigenetic mechanisms of SARS-CoV-2 infection, it is important to consider certain limitations. The generalizability of results may be influenced by the relatively small number of study participants, therefore larger cohorts are needed to validate our data. The limited sample size, especially in the follow-up analyses, poses certain challenges. The majority of 3 and 12-month follow-up samples were obtained from male participants with persisting respiratory symptoms, restricting our ability to account for potential sex-specific effects. To address this, we included an independent 6-month follow-up cohort that allowed us to assess symptom-dependent transcriptional changes. Although detailed baseline health information was unavailable for most participants due to the nature of the COVID-19 pandemic and the timing of recruitment, the independent cohort provided an opportunity to evaluate respiratory health relative to the pre-infection status.

Another limitation is the absence of mDNA-seq data of the follow-up samples due to a lack of available DNA. Thus, we were not able to analyze DNA methylation and gene expression levels concurrently in the follow-up samples, which limits the direct association to epigenetic changes. However, as mentioned above it is well-established that epigenetic reprogramming can lead to persistent alterations in gene expression. The genome-wide mDNA-seq was performed in a mixture of cells, and while we used the scRNA-seq data to discern the cells likely affected by the methylation changes, the specific cell types contributing to the COVID-19-related DNA methylation changes remain uncertain.

To our knowledge, this is the first study providing a comprehensive examination of the upper airway DNA methylome coupled with single-cell transcriptomics in the context of COVID-19. Our data reveal critical insights into SARS-CoV-2-induced DNA methylation changes in nasopharyngeal cells, representing the first airway barrier to be affected by the virus. Furthermore, we present preliminary evidence indicating a link between persisting transcriptional repression in epithelial cells and the presence of respiratory post-acute COVID-19 sequelae.

# Methods

### Reagents and tools table

| Reagent/resource | Reference or source | Identifier or catalog number |
|---|---|---|
| **Experimental models** | | |
| **Recombinant DNA** | | |
| **Antibodies** | | |
| **Oligonucleotides and other sequence-based reagents** | | |
| **Chemicals, enzymes, and other reagents** | | |
| DMEM/F12 medium | Gibco | 11039 |
| Dithiothreitol (DTT) | AppliChem | A2948 |
| Fetal bovine serum (FBS) | Gibco | 10500 |
| DMSO | Sigma-Aldrich | D8418 |

| Reagent/resource | Reference or source | Identifier or catalog number |
|---|---|---|
| PBS | Sigma-Aldrich | D8537 |
| Red Blood Cell Lysis Buffer | Roche | 11814389001 |
| Accutase | Thermo Fisher | 00-4555-56 |
| Chromium Next GEM Single Cell 3' Kits v3.1 | 10X Genomics | PN-1000121, PN-1000120, PN-1000123 |
| QIAamp® DNA Blood Mini Kit | QIAGEN | 51104 |
| NEBNext® Enzymatic Methyl-seq Kit (Protocol for Large Insert Libraries) | New England Biolabs | E7120 |
| **Software** | | |
| OTP pipeline | https://gitlab.com/one-touch-pipeline/otp | |
| bwa v.0.7.8 | https://github.com/lh3/bwa | |
| GATK | https://github.com/broadinstitute/gatk | |
| methylCtools v.1.0.0 | https://github.com/hovestadt/methylCtools | |
| EPISCORE | https://github.com/aet21/EpiSCORE | |
| DSS v.2.38 | https://github.com/haowulab/DSS | |
| BisSNP | https://github.com/dnaase/Bis-tools | |
| Cellranger v.3.0.1 | 10X Genomics | |
| Seurat v.3.2.2 | https://github.com/satijalab/seurat | |
| scanpy v.1.6.0 | https://github.com/scverse/scanpy | |
| harmony v.0.0.5 | https://github.com/immunogenomics/harmony | |
| bbknn v.1.4.0 | https://github.com/Teichlab/bbknn | |
| fgsea v.1.24.0 | https://github.com/alserglab/fgsea | |
| Cytoscape v.3.8.2 | https://cytoscape.org/ Shannon et al, 2003 | |
| **Other** | | |
| 35 μm cell strainer | Falcon | 352235 |
| Neubauer chamber | NanoEnTek | DHC-N01 |
| NovaSeq 6000 Sequencing System | Illumina | |
| ME220 Focused-Ultrasonicator | Covaris | |

## Study design and participants

Written informed consent was given by all patients or their legal representatives. Adult patients enrolled in either the prospective

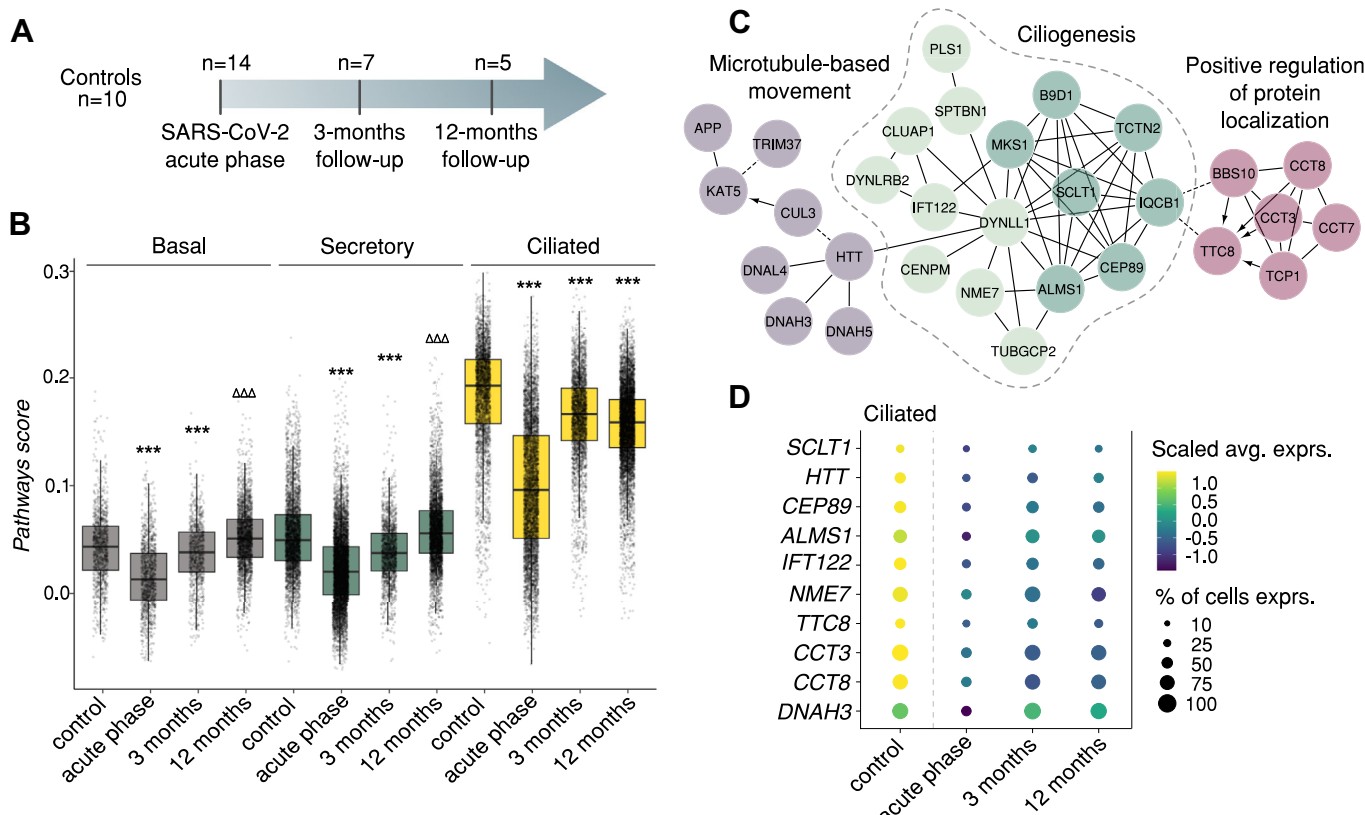

**Figure 4. Hypermethylation is associated with long-term repression of genes involved in ciliary function.**

(A) Postinfection follow-up samples were collected 3 and 12 months after infection and subjected to scRNA-seq. (B) Boxplot depicting *pathways score* levels (based on genes from Fig. 3A, lower panel) over time, groups were compared using pairwise Wilcoxon test, FDR-adj. *P* value < 0.001, higher than controls (ΔΔΔ) or lower than controls (***). The box represents the 25–75% quantiles, whiskers extend to 1.5 times of the interquartile range, and the median is indicated by a line (see Table EV5, for sample numbers, please see (A) (C) Cytoscape network of genes that remained repressed in ciliated cells comparing 12 month follow-up samples to controls (FDR-adj. *P* value < 0.05, MAST test adjusted for age and fraction of genes per cell, Dataset EV6). Module themes were assigned based on the most enriched pathways (FDR < 0.05, Table EV8). Dashed black lines: predicted protein interactions, solid lines: experimentally established protein interactions. Arrows indicate the directionality of interactions. (D) Dot plot showing example genes from the Cytoscape network (all genes are shown in Appendix Fig. S8). Circle color indicates expression strength and circle size represents the percentage of cells per group expressing the respective gene. All genes shown were significantly downregulated comparing acute phase (*n* = 14) vs. controls (*n* = 10) and 12 months (*n* = 5) vs. controls (FDR-adj. *P* value < 0.05, MAST test adjusted for age, fraction of genes per cell and sex, if applicable; see Dataset EV6).

observational cohort study Pa-COVID-19 at Charité–Universitätsmedizin Berlin (Kurth et al, 2020) or the SC2-study at the University of Leipzig Medical Center (Chua et al, 2020) were included in the analyses. Both studies were approved by the respective institutional review boards (Charité–Universitätsmedizin Berlin: EA2/066/20; University Hospital Leipzig: 123/20-ek) and were conducted in accordance with the Declaration of Helsinki. Nasopharyngeal swabs were collected from SARS-CoV-2-positive patients and controls and subjected to mDNA-seq and scRNA-seq. Samples were obtained from March to May, 2020, during the first wave of the COVID-19 pandemic in Germany (SARS-CoV-2 wild-type). The presence or absence of SARS-CoV-2 RNA was confirmed by real-time PCR. Controls tested negative for SARS-CoV-2 and had not been vaccinated against SARS-CoV-2. Those patients who attended follow-up visits at 3 and 12 months post infection were resampled for scRNA-seq. In addition, independent follow-up samples were collected 6 months post infection as part of the SC2-study from former non-hospitalized COVID-19 patients with mild acute disease. The persistence of respiratory symptoms was assessed using the modified Medical Research Council (mMRC) dyspnea scale (Mahler et al, 1987). If the

mMRC grade was ≥1, the patient was classified as having persistent respiratory symptoms. No formal sample size calculation was performed; the sample size was determined based on maximal availability. Data collection and analysis were not performed blind to the conditions of the experiments.

## Isolation of cells from nasal swabs and single-cell 3′ RNA library preparation

Nasopharyngeal swabs were collected from all COVID-19 patients that were hospitalized in Charité – Universitätsmedizin Berlin or the University Hospital Leipzig from March to May 2020 that were willing to participate in this study. In addition, nasopharyngeal swabs were collected 6 months post infection from former non-hospitalized COVID-19 patients with mild acute disease. Controls were individuals hospitalized for orthopedic/esthetic surgery and tested negative for SARS-CoV-2. Cells used for single-cell RNA and whole-genome methylation sequencing were isolated from freshly collected nasopharyngeal swabs and processed in analogy to

previous publications (Chua et al, 2020; Loske et al, 2022; Trump et al, 2021). Following sampling, the swabs were directly transferred into 500 μL precooled DMEM/F12 medium (Gibco), kept on ice and processed further within one hour after collection. Subsequently, 500 μL of 13 mM DTT (AppliChem) were added to the nasopharyngeal swabs. In order to dissociate cells, the suspension was mixed by pipetting cautiously against the swab, followed by dipping the swab 20 times into the medium. Samples were then incubated in a thermomixer (37 °C, 500 rpm, 10 min), spun down (350×g, 4 °C, 5 min) and the supernatant removed without disturbing the cell pellet. In some cases, at this stage, cell pellets were resuspended in cryopreservation medium [20% fetal bovine serum (FBS, Gibco), 10% DMSO (Sigma-Aldrich), 70% DMEM/F12] and stored at −80 °C. To continue the protocol with frozen cells, these were quickly thawed at 37 °C, spun down (350×g, 4 °C, 5 min), and processed as described in the following.

If blood traces were visible, cell pellets were resuspended in 500 μL PBS (Sigma-Aldrich) and treated with 1 mL of RBC Lysis Buffer (Roche) at 25 °C for 10 min and spun down thereafter (350×g, 4 °C, 5 min). For all samples, 500 μL Accutase (Thermo Fisher) were added to the cell pellets for a 10-min incubation at room temperature. To terminate the enzymatic digestion, 500 μL DMEM/F12 with 10% FBS were added, and cells were spun down (350×g, 4 °C, 5 min). Next, pellets were resuspended in 100 μL 1× PBS and filtered through a 35 μm cell strainer (Falcon) to ensure the removal of cell clumps. Cells were counted using a disposable Neubauer chamber (NanoEnTek). Single-cell suspensions and RNA libraries were generated with the Chromium Next GEM Single Cell 3' Kits v3.1 (10X Genomics). For the single-cell suspension, a master mix containing 17,500 cells per sample was loaded onto the 10X Chromium Controller. The remaining cells were either used directly for DNA isolation or pelleted and stored at −80 °C until further use. The generation of gel bead emulsions together with the subsequent steps of reverse transcription, cDNA amplification, and library preparation were performed as described in the manufacturer's protocol. The incubation at 85 °C during cDNA synthesis was prolonged by 5 min to ensure virus inactivation. Final 3' RNA libraries were combined to pools of eight or up to 24 samples (S2 or S4 flow cell, respectively) and sequenced with the NovaSeq 6000 Sequencing System (Illumina; paired-end, single indexing).

## Whole-genome DNA methylation sequencing (mDNA-seq)

DNA was isolated from human airway-derived cell pellets using the QIAamp® DNA Blood Mini Kit (QIAGEN), following the instructions of the Blood or Body Fluid Spin Protocol. Up to 150 ng of isolated DNA were fragmented to an average length of 350–400 bp using the ME220 Focused-Ultrasonicator (Covaris). DNA libraries were prepared with the NEBNext® Enzymatic Methyl-seq Kit (New England Biolabs). In brief, DNA was first end-repaired, dA-tailed, and adapter-ligated. In the next step, 5-methylcytosines and 5-hydroxymethylcytosines were oxidized by the TET2 enzyme. The oxidized cytosine derivatives evade the subsequent deamination reaction, during which APOBEC deaminates cytosines to uracils. Finally, libraries were amplified using unique dual-index primers. All samples were sequenced using the NovaSeq 6000 Sequencing System (Illumina; 150-bp paired-end) in pools of 12 samples on S4 flow cells.

## Processing and analysis of whole-genome methylation sequencing data

Whole-genome methylation data were pre-processed by the OTP pipeline (Reisinger et al, 2017). In particular, adapter sequences were clipped, and paired-end reads were mapped to the human genome (assembly GRCh37/hg19) with bwa v.0.7.8. Duplicate reads were identified using the *MarkDuplicates* tool from GATK (Van der Auwera and O'Connor, 2020) and removed. Subsequently, methylation calling for each cytosine in the human genome was performed using methylCtools v.1.0.0 (Hovestadt et al, 2014) which also calculates a probability score for a cytosine being a single-nucleotide polymorphism (SNP). For the whole cohort, we excluded all CpGs from downstream analysis if the SNP score in any of the samples was greater or equal to 0.25 or if the CpG position had a coverage of less than 5X in all samples. In addition, we filtered out high-mappability regions from ENCODE (Amemiya et al, 2019).

Partially methylated domains (PMDs) were identified for each sample using MethylSeekR (Burger et al, 2013). As input, chromosome-wide methylation count matrices were generated, including the total read coverage and the number of methylated cytosines for each CpG position. After PMDs were identified, the remaining regions were further subdivided into lowly methylated (LMRs) and unmethylated regions by using the the *segmentUMRsLMRs* function. LMRs, i.e., putative regulatory regions, were used to determine differentially methylated regions (DMRs) using DSS v.2.38 (Wu et al, 2015). As input, we used a summarized LMR methylation value, calculated by averaging the methylation levels of the all CpGs with a single-nucleotide polymorphism (SNP) probability score below 0.25, as determined by methylCtools v.1.0.0. within each LMR (Hovestadt et al, 2014). DMRs between SARS-CoV-2-positive ($n = 19$) and negative ($n = 14$) samples were determined while adjusting for participants' age, sex, and cell lineage composition (Table EV1). Where available, cell compositions were derived from the corresponding scRNA-seq data. In case no matching scRNA-seq data were available, cellular composition was calculated using EPISCORE (Teschendorff et al, 2020), which constructs a sample-specific DNA methylation reference matrix based on the scRNA-seq data from the matched samples, and subsequently uses this matrix for inferring the cellular composition for the unmatched samples based on their methylation values. If an LMR region was flagged as differentially methylated and the average CpG coverage was greater or equal 10X, we reported the LMR together with the FDR-adjusted $p$ value from the DSS Wald-test. DMRs were screened for overlaps with enhancers from GeneHancer v.4.8 (Fishilevich et al, 2017), ENCODE (Encode Project Consortium, 2012), and Roadmap (15-state model, enhancers identified in primary cells of the Blood & T-cell or HSC & B-cell groups, (Roadmap Epigenomics Consortium et al, 2015) databases. Furthermore, DMR overlaps with promoters and gene bodies were determined based on GENCODE v19 annotation (Frankish et al, 2019). If a DMR overlapped with an enhancer, the DMR's genomic feature was classified as such. If no overlapping enhancer was found, the DMR was screened for promoters, followed by gene bodies, and annotated accordingly. DMRs for which no overlapping element was found, were classified as *intergenic*. DMR target genes were assigned as follows: In case the DMR overlapped with GeneHancer v.4.8 enhancers (Fishilevich

et al, 2017), the respective enhancer target genes were annotated. If no overlapping enhancer was found, the gene with the closest TSS to the DMR was annotated, provided the distance was within 1 million bp. SNPs in each sample were identified using BisSNP (Liu et al, 2012) and used to determine methylation quantitative trait loci (meQTL), i.e., genetic variants that may affect the methylation of surrounding CpG sites. To this end, all identified SNPs in a 5 kb genomic region around each DMR were tested for association with the average DMR methylation. In particular, we determined if a SNP was present on one or both alleles for each of the 33 samples in the cohort and performed a Spearman correlation test by using the *cor.test* function in R. For a SNP considered for testing we demanded a minimum of 3 out of the 33 samples to have the SNP called and not called. The obtained *p*-values were FDR-adjusted and a significant SNP-DMR association (methylation quantitative trait loci, meQTL) was inferred if the adjusted $P$ value was less than 0.1. In case a meQTL was detected, the corresponding DMR was classified as genotype-associated (gDMR), DMRs with no significant SNP association were termed ngDMRs (non-genotype-associated). A heatmap of the average methylation values per DMR across the 33 samples was generated by using the Complex-Heatmap package (Gu et al, 2016) in R. Rows and columns were clustered individually by using the ward.D2 method in the *hclust* function.

## Processing and analysis of single-cell 3′ RNA sequencing data

Cellranger v.3.0.1 was used for processing the scRNA-seq data. For read alignment, the hg19 reference genome (10x genomics, version 3.1.0) was extended by the SARS-CoV-2 genome (Refseq-ID: NC_045512), which was added as an additional chromosome. Cellranger output files were further processed using Seurat v.3.2.2 (Stuart et al, 2019). Genes expressed in at least three different cells were retained and only cells containing 200 or more genes with a mitochondrial reads percentage of less than 15% were used for further analyses. Counts of Unique Molecular Identifiers were then plotted against gene counts per cell for each sample and an upper gene count cut-off was determined visually in order to exclude outliers. All samples were merged into one object, metadata and counts were exported and loaded into Scanpy v.1.6.0 (Wolf et al, 2018) for further processing. Counts were normalized to 10,000 reads per cell and log-transformed. Highly variable genes were identified and used for subsequent sample integration. The scaled and PCA transformed data were integrated using harmony v.0.0.5 (Korsunsky et al, 2019) and bbknn v.1.4.0, (Polanski et al, 2020) based on 80 principal components. Next, UMAP embedding and Leiden clusters (Traag et al, 2019) were calculated and Leiden cluster resolution of 1.1 was used to annotate different cell types and states based on marker gene expression (Appendix Fig. S1). Macrophage/dendritic cell, T-cell and Secretory clusters were further subclustered for cell type identification. Finally, the object was converted to a h5seurat file using SeuratDisk v.0.0.0.9014 and loaded into R v.4.2.3 (R Core Team, 2022) where subsequent analyses were performed using Seurat v.4.3.0.1 (Butler et al, 2018).

Differentially expressed DMR target genes between SARS-CoV-2-positive samples and controls were calculated using the *FindMarkers* function (logfc.threshold=0.01, min.pct=0.05) in Seurat under the use of the MAST test. MAST is based a hurdle

model adapted to scRNA-seq data and allows for the consideration of covariates, which were set to participants' age, the proportion of genes detected within each cell (cellular detection rate, see (Finak et al, 2015)) and sex, where applicable. For correlating DNA methylation with gene expression, pseudo bulk expression per cell type and sample was calculated using the *AverageExpression* function in Seurat and transformed back into log-space using *log1p*. Two-tailed Spearman correlation between average methylation at the hypoDMR on chr3p21 and pseudo bulk expression of all enhancer target genes was calculated using the *cor.test* function in R. Gene set enrichment analysis was performed with fgsea v.1.24.0 (Korotkevich et al, 2021) under the use of Gene Ontology gene sets (c5.all.v2023.1.Hs.symbols.gmt). Only the top six most significant pathways with positive or negative enrichment scores were considered. The *pathways score* for repressed pathway genes from the gene set enrichment analysis ($n = 323$) as well as the *hypoDMR network score* were calculated for each cell using the *ModuleScore* function in Seurat, which determines a score by averaging expression levels of a given gene set and subtracting the expression levels of a randomly selected control gene set of similar expression strength. The p*athways score* was visualized as a *FeaturePlot*, downsampled to 9000 cells per group using the *subset* function. For all *pathways score*-related analyses, epithelial cells were grouped into basal cells = Basal, early_progenitor, Basal_diff; secretory cells = Club, Goblet, Sec_diff and ciliated cells = Ciliated, Cil_diff, SAA_rich_Cil cells (see Appendix Fig. S2). Cellular proportion shifts were determined with scCODA (Büttner et al, 2021) under use of the AnnData object in scanpy, with mDCs set as the reference cell type. For comparisons, the object was subset to the two groups of interest, and credible effects (FDR-adj. $P < 0.05$) were extracted using the *result.effect_df* function. Differences in the *pathways score* between groups were assessed by pairwise Wilcoxon test in R. Differentially expressed *pathways score* genes between 12 month follow-up samples and controls were calculated as described above, except that sex could not be used as a covariate, as the 12 month follow-up samples were derived from male participants only. Differentially expressed pathway genes were visualized using the *DotPlot* function (Seurat). Multiple linear regression for the 6 month follow-up samples was performed with the *lm* function in R while adjusting for age, sex and fraction of genes detected within each cell.

## Promoter Capture Hi-C interaction data

Promoter Capture Hi-C interaction data were retrieved from supplementary data S1 from Javierre et al, 2016 (Javierre et al, 2016). In particular, promoter-enhancer regions from file PCHiC_-peak_matrix_cutoff5.tsv were scanned for overlaps with our DMR regions using intersectBed. An interaction plot was created with the Python package gelviz v.0.9 (Bieg, 2021).

## Gene network analyses

Hypo DMR target genes were subjected to Enrichr-KG (Evangelista et al, 2023) and screened for overrepresentation in Gene Ontology pathways. To assess the function of differentially expressed *pathways score* genes comparing 12 month follow-up samples to controls in the ciliated cell group, we used Cytoscape v.3.8.2 (Shannon et al, 2003) with the ReactomeFIPlugIn v.2021 (Wu et al, 2014). All genes that

**The paper explained**

**Problem**

The pathomechanisms of post-COVID-19 condition remain poorly understood. As DNA methylation is considered a relatively stable epigenetic modification, its alterations may affect gene expression beyond the acute phase, possibly contributing to the persistence of post-acute sequelae of COVID-19.

**Results**

We detected a large number of hypermethylated regions in COVID-19 patients that were linked to genes involved in ciliary function. The corresponding genes were not only downregulated in the acute phase of SARS-CoV-2 infection but also 3 and 12 months post infection, suggesting that the hypermethylation observed in the acute phase of infection may exert a long-term impact on gene expression. These findings were replicated in an independent cohort collected 6 months post infection.

**Impact**

The data suggest that respiratory post-acute sequelae of COVID-19 may be related to long-term transcriptional dysregulation in airway epithelial cells, likely associated with epigenetic perturbations initiated in the acute phase of the disease. This could help to identify potential therapeutic targets for managing post-COVID-19 conditions.

were repressed in the acute phase of infection as well as 12 months post infection in ciliated cells were loaded into the *Gene Set/Mutation Analysis* tool, and the network was constructed without linker genes. After module clustering, enrichment for GO Biological Process pathways was performed considering all modules containing at least three genes (FDR-adj. $P < 0.05$).

## Motif enrichment analysis

Motif enrichment was performed using the SEA tool (Bailey and Grant, 2021) in MEME suite v.5.5.5 under the use of the HOCOMOCO database v.11. Motif abundancies within those hyperDMRs that were related to continuously repressed genes in ciliated cells were compared to motives found in all other hyperDMRs, and only those motives with an FDR-adjusted $P$ value < 0.05 were considered.

## Data availability

The scRNA-seq data for part of the subjects analyzed in acute COVID-19 have been previously published (Chua et al, 2020) and are available in the European Genome-phenome Archive repository under EGAS00001004481. Due to the potential risk of re-identification of pseudonymized raw sequencing data, the counts of total and methylated reads for all SNP-filtered CpG sites per sample (as used for the DMR calling) as well as the FASTQ files for further scRNA-seq data can be obtained through EGAS50000000273 for non-commercial research purposes alone, subject to controlled access as mandated by EU data protection laws. For access, contact the corresponding author, STr. Processed data in the form of a Seurat object are available on FigShare for the

independent 6-month follow-up samples (https://doi.org/10.6084/m9.figshare.28485914).

The source data of this paper are collected in the following database record: biostudies:S-SCDT-10_1038-S44321-025-00215-5.

## Peer review information

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

## Acknowledgements

First and foremost, we are grateful to the participants of the Pa-COVID-19 and the SC2-studies. The authors thank Illumina GmbH for financial support via the allocation of reagents and sequencing flow cells, as well as Markus Vossmann, Martin Allgaier, and Oliver Krätke for the realization of the sequencing runs at the Illumina Solutions Center Berlin. The authors would further like to thank Philipp Strubel and Stefan Schneider from the BIH Center of Digital Health for their support in data management as well as the BIH Core Unit Genomics for performing sequencing runs. Lastly, the authors wish to thank Intel Germany GmbH for generously donating computational infrastructure. This study was supported by the BIH COVID-19 research program, the European commission (ESPACE, 874719, Horizon 2020), the BMBF-funded de.NBI Cloud within the German Network for Bioinformatics Infrastructure (de.NBI; 031A537B, 031A533A, 031A538A, 031A533B, 031A535A, 031A537C, 031A534A, 031A532B), the BMBF-funded Medical Informatics Initiative (HiGHmed, 01ZZ1802A-01ZZ1802Z; Calm-QE 01ZZ2318G), the BMBF-funded SAGE project (031L0265) and the EU ISIDORe PATH2XNAT project (ISID_JRA_f19x). The synopsis figure was created with *Biorender.com*.

## Author contributions

**Marey Messingschlager**: Conceptualization; Formal analysis; Investigation; Visualization; Writing—original draft; Writing—review and editing. **Sebastian D Mackowiak**: Conceptualization; Formal analysis; Writing—original draft. **Maria Theresa Voelker**: Conceptualization; Resources; Writing—original draft. **Matthias Bieg**: Formal analysis. **Jennifer Loske**: Conceptualization; Formal analysis; Investigation. **Robert Lorenz Chua**: Investigation. **Johannes Liebig**: Investigation. **Sören Lukassen**: Conceptualization; Resources; Formal analysis. **Loreen Thürmann**: Conceptualization. **Anke Seegebarth**: Investigation. **Sven Twardziok**: Data curation. **Daria Doncevic**: Formal analysis. **Carl Herrmann**: Conceptualization. **Stephan Lorenz**: Resources; Investigation. **Sven Klages**: Resources; Investigation. **Fridolin Steinbeis**: Resources. **Martin Witzenrath**: Resources. **Florian Kurth**: Resources. **Christian Conrad**: Conceptualization. **Leif E Sander**: Resources. **Naveed Ishaque**: Conceptualization; Resources; Supervision. **Roland Eils**: Conceptualization; Supervision; Funding acquisition. **Irina Lehmann**: Conceptualization; Supervision; Funding acquisition; Writing—review and editing. **Sven Laudi**: Conceptualization; Resources; Supervision; Writing—review and editing. **Saskia Trump**: Conceptualization; Supervision; Visualization; Writing—original draft; Writing—review and editing.

Source data underlying figure panels in this paper may have individual authorship assigned. Where available, figure panel/source data authorship is listed in the following database record: biostudies:S-SCDT-10_1038-S44321-025-00215-5.

## Funding

## Disclosure and competing interests statement

The authors declare no competing interests.

