## [Peer Review File · EMBO Molecular Medicine]

DNA Methylation Changes in COVID-19 Link to Long-Term Transcriptional Dysregulation in Airway Cells

Marey Messingschlager, Sebastian Mackowiak, Maria Völker, Matthias Bieg, Jennifer Loske, Robert Chua, Johannes Liebig, Sören Lukassen, Loreen Thürmann, Anke Seegebarth, Sven Twardziok, Daria Doncevic, Carl Herrmann, Stephan Lorenz, Sven Klages, Fridolin Steinbeis, Martin Witzenrath, Florian Kurth, Christian Conrad, Leif Sander, Naveed Ishaque, Roland Eils, Irina Lehmann, Sven Laudi, and Saskia Trump

Corresponding author: Saskia Trump (saskia.trump@charite.de)

Review Timeline:

Submission Date:	30th Jan 25
Editorial Decision:	19th Feb 25
Revision Received:	28th Feb 25
Accepted:	3rd Mar 25

Editor: Jingyi Hou

Transaction Report:

(Note: Please note that the manuscript was previously reviewed at another journal and the reports were taken into account in the decision making process at EMBO Molecular Medicine. Since the original reviews are not subject to EMBO Press' transparent review process policy, the reports and author response cannot be published. With the exception of the correction of typographical or spelling errors that could be a source of ambiguity, letters and reports are not edited. Depending on transfer agreements, referee reports obtained elsewhere may or may not be included in this compilation. Referee reports are anonymous unless the Referee chooses to sign their reports.)

19th Feb 2025

Dear Dr. Trump,

Thank you for submitting your manuscript to EMBO Molecular Medicine. We have now received the enclosed report from the referee who was asked to assess your work and your response to the reviewers comments from the previous journal. As you will see, the referee thinks the raised concerns have been addressed, and I am pleased to inform you that we will be able to accept your manuscript pending the following amendments:

1. We suggest changing "Dysregulation of Transcriptional Programs" to "Transcriptional Dysregulation" in the title.
2. Please provide a written response to the reviewer's comments regarding the lack of definitive proof of long-term epigenetic modifications in a 'point-by-point response' and discuss this as a limitation.

On a more editorial level, please address the following issues:

1. Main figures should be removed from the manuscript and uploaded as individual, high resolution figure files. The legends should be compiled at the end of the manuscript text.
2. Please provide a complete author checklist, which you can download from our author guidelines (<https://www.embopress.org/page/journal/17574684/authorguide#submissionofrevisions>). Please insert information in the checklist that is also reflected in the manuscript. The completed author checklist will also be part of the Review Process File (RPF).
3. At EMBO Press we ask authors to provide source data for the main manuscript figures. Our source data coordinator has already contacted you to discuss which figure panels we would need source data for and provided you with helpful tips on how to upload and organize the files.
4. Funding information should be merged with Acknowledgements. Complete funding information should be entered into our submission system as well.
5. Appendix: the "Online Methods" (and References) should be removed from the file and added to the "Methods" in the main manuscript text. The supplementary figures should be renamed to "Appendix Figure S1 - S10" and their callouts should also be corrected in the main manuscript text. The files should be uploaded as a PDF, labelled "Appendix", and a table of contents with page numbers should be added to the first page. Please remove the track changes in the Appendix file.
6. Supplementary tables: Table S3, S4, S6, S7, S9, S13, S18 should be renamed to "Dataset EV1" - 7 and uploaded as separate files. Tables S1, S2, S5, S8, S10, S11, S12, S14, S15, S16, S17 should be renamed to "Table EV1" - 11 and uploaded as separate files. The callouts should be updated accordingly.
7. The references need to be formatted according to the EMBO Molecular Medicine reference style. Please list up to 10 co-authors of a paper before adding et al. in the reference list. Citations should be listed in alphabetical order.
8. The paper explained: EMBO Molecular Medicine articles are accompanied by a summary of the articles to emphasize the major findings in the paper and their medical implications for the non-specialist reader. Please provide a draft summary of your article highlighting
 - the medical issue you are addressing,
 - the results obtained and
 - their clinical impact.

9. Author contributions: You will be asked to provide CRediT (Contributor Role Taxonomy) terms in the submission system. These replace a narrative author contribution section in the manuscript. Please remove the Author contribution section from the manuscript.
10. Every published paper includes a 'Synopsis' to further enhance discoverability. Synopses are displayed on the journal webpage and are freely accessible to all readers. They include a short stand first (maximum of 300 characters, including space) as well as 2-5 one-sentences bullet points that summarizes the paper. Please write the bullet points to summarize the key NEW findings. They should be designed to be complementary to the abstract - i.e. not repeat the same text. We encourage inclusion of key acronyms and quantitative information (maximum of 30 words / bullet point). Please use the passive voice. Please attach

these in a separate file or send them by email, we will incorporate them accordingly.

Please also suggest a visual abstract to illustrate your article as a PNG file 550 px wide x 300-600 px high.

12. All Materials and Methods need to be described in the main text using our 'Structured Methods' format. According to this format, the Methods section includes a Reagents and Tools Table (listing key reagents, experimental models, software and relevant equipment and including their sources and relevant identifiers) followed by a Methods and Protocols section describing the methods, ideally using a step-by-step protocol format. The aim is to facilitate adoption of the methodologies across labs.

Please download and fill our Reagents and Tools Table template (.docx), which you can find in our author guidelines:
<https://www.embopress.org/page/journal/17574684/authorguide#structuredmethods>

When submitting your revised manuscript, please DO NOT include the Reagents and Tools Table in the Methods section of the manuscript but upload it as a separate file choosing the file type "Reagent Table".

11. Please use the following order of the manuscript sections: Abstract / Keywords / The Paper Explained / Introduction / Results / Discussion / Methods / Data Availability / Acknowledgements / Disclosure and Competing Interests Statement / References / Main Figure Legends / Tables / Expanded View Figure Legends

12. The panel label "D" is missing the figure file of Fig2D.

13. Remove the "take-home message".

14. Data availability: please add the EGA ID.

15. Please address the following issues related to Figure legends:

- Please note that the exact p values are not provided in the legends of figures 4B
- Please note that the box plots need to be defined in terms of minima, maxima, centre, in the legends of figures 4B
- Please note that information related to n is missing in the legends of figures 2B, 4B

To submit your manuscript, please follow this link:

<https://embomolmed.msubmit.net/cgi-bin/main.plex>

I look forward to receiving a revised version of your manuscript as soon as possible.

Kind regards,
Jingyi

Jingyi Hou
Senior Editor
EMBO Molecular Medicine

*** Instructions to submit your revised manuscript ***

- 1) a .docx formatted version of the manuscript text (including Figure legends and tables)
- 2) Separate figure files*
- 3) supplemental information as Expanded View and/or Appendix. Please carefully check the authors guidelines for formatting Expanded view and Appendix figures and tables at <https://www.embopress.org/page/journal/17574684/authorguide#expandedview>
- 4) a letter INCLUDING the reviewer's reports and your detailed responses to their comments (as Word file).
- 5) The paper explained: EMBO Molecular Medicine articles are accompanied by a summary of the articles to emphasize the major findings in the paper and their medical implications for the non-specialist reader. Please provide a draft summary of your article highlighting
 - the medical issue you are addressing,
 - the results obtained and
 - their clinical impact.This may be edited to ensure that readers understand the significance and context of the research. Please refer to any of our published articles for an example.
- 6) Author contributions: the contribution of every author must be detailed in a separate section.
- 7) EMBO Molecular Medicine now requires a complete author checklist (<https://www.embopress.org/page/journal/17574684/authorguide>) to be submitted with all revised manuscripts. Please use the checklist as guideline for the sort of information we need WITHIN the manuscript. The checklist should only be filled with page numbers where the information can be found. This is particularly important for animal reporting, antibody dilutions (missing) and exact values and n that should be indicated instead of a range.

8) Every published paper now includes a 'Synopsis' to further enhance discoverability. Synopses are displayed on the journal webpage and are freely accessible to all readers. They include a short stand first (maximum of 300 characters, including space) as well as 2-5 one sentence bullet points that summarise the paper. Please write the bullet points to summarise the key NEW findings. They should be designed to be complementary to the abstract - i.e. not repeat the same text. We encourage inclusion of key acronyms and quantitative information (maximum of 30 words / bullet point). Please use the passive voice. Please attach these in a separate file or send them by email, we will incorporate them accordingly.

You are also welcome to suggest a striking image or visual abstract to illustrate your article. If you do please provide a jpeg file 550 px-wide x 300-600px high.

- 9) A Conflict of Interest statement should be provided in the main text
- 10) Please note that we now mandate that all corresponding authors list an ORCID digital identifier. This takes <90 seconds to complete. We encourage all authors to supply an ORCID identifier, which will be linked to their name for unambiguous name identification.

Currently, our records indicate that the ORCID for your account is 0000-0002-9894-1807.

Link Not Available

- 11) Include a Reagents and Tools Table as part of the Methods section, which can be downloaded from our author guidelines (<https://www.embopress.org/page/journal/17574684/authorguide#structuredmethods>)

Photos 400-800 DPI

*Additional important information regarding figures and illustrations can be found at <https://bit.ly/EMBOPressFigurePreparationGuideline>. See also figure legend preparation guidelines: <https://www.embopress.org/page/journal/17574684/authorguide#figureformat>

***** Reviewer's comments *****

Referee #1 (Comments on Novelty/Model System for Author):

The manuscript of Messingschlager et al. investigates transcriptomic and epigenetic (DNA Methylation) in samples of the upper respiratory tract after SARS-CoV-2 infection.

The manuscript was submitted to another journal and evaluated by two referees which provided suggestions for improvement.

The authors revised their manuscript and wrote a rebuttal letter both of which were now submitted to EMBO molecular medicine.

The methods and statistical analyses of the data are sound.

A similar longitudinal study with samples of the upper respiratory tract has not been performed (to my knowledge).

There is no mechanistic data, thus the medical impact is difficult to assess.

Model system is human.

Referee #1 (Comments for Author):

I understand that the editor is asking for my opinion on the manuscript in general and the extent to which the authors have addressed the critiques raised by the two referees.

My general comments are listed above. The study is interesting and provides a starting point for a better understanding of post-COVID conditions. It is based exclusively on genomic sequencing data, with no accompanying phenotypic or mechanistic data. However, a novel follow-up cohort was introduced, in which transcriptomic data were correlated with clinical findings.

Overall, the quantity of data added in response to reviewer comments is extensive, and the quality is good. The authors were asked to tone down their interpretations, as there were many overstatements regarding mechanistic findings. This issue has been addressed.

The authors also claim that the DNA methylation observed in this study affects immune regulatory genes and genes associated with ciliary function, with these changes persisting 12 months post-infection. However, the study fails to link these transcriptomic changes with long-term epigenetic modifications, as suggested by the referees. This is surprising, as the quantification of DNA methylation is technically straightforward.

In conclusion, the majority of the critiques raised by the referees have been addressed. However, one key point-providing definitive proof of long-term epigenetic modifications is still missing.

***** Reviewer's comments *****

Referee #1 (Comments on Novelty/Model System for Author):

The manuscript of Messingschlager et al. investigates transcriptomic and epigenetic (DNA Methylation) in samples of the upper respiratory tract after SARS-CoV-2 infection.

The manuscript was submitted to another journal and evaluated by two referees which provided suggestions for improvement.

The authors revised their manuscript and wrote a rebuttal letter both of which were now submitted to EMBO molecular medicine.

The methods and statistical analyses of the data are sound.

A similar longitudinal study with samples of the upper respiratory tract has not been performed (to my knowledge).

There is no mechanistic data, thus the medical impact is difficult to assess.

Model system is human.

Referee #1 (Comments for Author):

I understand that the editor is asking for my opinion on the manuscript in general and the extent to which the authors have addressed the critiques raised by the two referees.

My general comments are listed above. The study is interesting and provides a starting point for a better understanding of post-COVID conditions. It is based exclusively on genomic sequencing data, with no accompanying phenotypic or mechanistic data. However, a novel follow-up cohort was introduced, in which transcriptomic data were correlated with clinical findings.

Overall, the quantity of data added in response to reviewer comments is extensive, and the quality is good. The authors were asked to tone down their interpretations, as there were many overstatements regarding mechanistic findings. This issue has been addressed.

The authors also claim that the DNA methylation observed in this study affects immune regulatory genes and genes associated with ciliary function, with these changes persisting 12 months post-infection. However, the

study fails to link these transcriptomic changes with long-term epigenetic modifications, as suggested by the referees. This is surprising, as the quantification of DNA methylation is technically straightforward.

In conclusion, the majority of the critiques raised by the referees have been addressed. However, one key point-providing definitive proof of long-term epigenetic modifications is still missing.

Thank you for your comment. As described in the manuscript, we were unable to assess DNA methylation levels in the follow-up samples due to the unavailability of remaining cells or DNA from these samples. We acknowledge that linking long-term transcriptomic changes to persistent epigenetic modifications would have been valuable. However, given this technical constraint, such an analysis was not feasible in our study.

To ensure that this limitation is more clearly stated, we have now emphasized it in the discussion (lines 377–379, new text in bold):

“Another limitation is the absence of mDNA-seq data of the follow-up samples due to a lack of available DNA. Thus, we were not able to analyse DNA methylation and gene expression levels concurrently in the follow-up samples, which limits the direct association to epigenetic changes.”

Furthermore, in the *Paper explained* section we slightly shifted the focus to the persistent transcriptional dysregulation while acknowledging that these changes are most likely associated with epigenetic perturbations initiated during the acute phase of the disease:

“The data suggest that respiratory post-acute sequelae of COVID-19 may be related to long-term transcriptional dysregulation in airway epithelial cells, likely associated to epigenetic perturbations initiated in the acute phase of disease.”

3rd Mar 2025

Dear Saskia,

Thank you for sending us your revised manuscript. I am pleased to inform you that your manuscript is accepted for publication and is now being sent to our publisher to be included in the next available issue of EMBO Molecular Medicine.

Yours sincerely,
Jingyi

Jingyi Hou
Senior Editor
EMBO Molecular Medicine
